# Advances in *Babesia* Vaccine Development: An Overview

**DOI:** 10.3390/pathogens12020300

**Published:** 2023-02-11

**Authors:** Michał Jerzak, Albert Gandurski, Marta Tokaj, Weronika Stachera, Magdalena Szuba, Monika Dybicz

**Affiliations:** Department of General Biology and Parasitology, Medical University of Warsaw, 02-004 Warsaw, Poland

**Keywords:** vaccine, babesiosis, *Babesia microti*, *B. bovis*, *B. bigemina*, *B. divergens*, *B. orientalis*

## Abstract

Babesiosis is a tick-borne zoonotic disease, which is caused by various species of intracellular *Babesia* parasite. It is a problem not only for the livestock industry but also for global health. Significant global economic losses, in particular in cattle production, have been observed. Since the current preventive measures against babesiosis are insufficient, there is increasing pressure to develop a vaccine. In this review, we survey the achievements and recent advances in the creation of antibabesiosis vaccine. The scope of this review includes the development of a vaccine against *B. microti*, *B. bovis*, *B. bigemina*, *B. orientalis* and *B. divergens*. Here, we present different strategies in their progress and evaluation. Scientists worldwide are still trying to find new targets for a vaccine that would not only reduce symptoms among animals but also prevent the further spread of the disease. Molecular candidates for the production of a vaccine against various *Babesia* spp. are presented. Our study also describes the current prospects of vaccine evolution for successful *Babesia* parasites elimination.

## 1. Introduction

*Babesia* species belong to apicomplexan hemoprotozoan parasites of the genus *Babesia* [1,2]. They are intraerythrocytic and obligate intracellular parasites [2,3,4]. Victor Babes was the first to discover those parasites. He came across them in 1888 while investigating cattle herds diagnosed with hemoglobinuria and initially named them *Haematococcus bovis* [5]. Nowadays, we have reported over 100 *Babesia* spp., which are suspected to be the second most common blood parasite (after trypanosomes) [2,6]. The infection of both animals and humans may lead to a disease known as babesiosis, a tick being the common vector all around the world [3,4]. In humans, the transmission pathways include not only ticks but also blood transfusion, a perinatal route through the placenta and organ transplantation [7,8,9,10,11]. Currently, babesiosis is classified as an emerging zoonotic disease. *Babesia microti* is commonly found in Japan, United States of America and northeastern Eurasia [12,13,14].

The complex life cycle of the parasite consists of the development of asexual stages in vertebrate hosts and the sexual ones in ticks [1,15,16] (Figure 1).

The asexual stage specifically takes place inside red blood cells (RBCs) [1]. Infected ticks introduce the protozoans when feeding on the blood of the mammalian host. The infection starts when sporozoites of *Babesia* spp. enter erythrocytes and multiply (asexual reproduction). Some parasites transform into male and female gametocytes. Once the tick takes up some blood containing gametocytes, the sexual stage begins. The zygote undergoes sporogony, which results in the production of sporozoites [17]. 

Despite the fact that babesiosis is found primarily in animals, the prevalence of human babesiosis has been recently increasing. Symptoms include increased temperature, anemia, hemoglobinuria, jaundice and splenomegaly [6]. Human babesiosis is caused mostly by *B. microti* in the USA and *B. divergens* in Europe [16,18]. Clinical symptoms of human babesiosis range from mild to severe. Immunocompetent patients are mainly asymptomatic or have mild, flu-like symptoms. Acute babesiosis is characterized by fever, hemolytic anemia, hemoglobinuria and occasionally death. In immunocompromised persons, babesiosis is characterized by renal failure, acute respiratory distress and disseminated intravascular coagulations [4]. Life-threatening complications can occur in immunocompromised patients, for example, people without spleen, with cancer history, individuals with AIDS, neonates, the elderly and people taking immunosuppressive drugs [19,20,21]. Complications include acute respiratory distress syndrome (ARDS), severe anemia, congestive heart failure, disseminated intravascular coagulopathy (DIC), liver and renal failure and shock [20,22,23]. Unfortunately, there are no currently available vaccines to protect humans against babesiosis. Babesiosis is prevalent among many mammals, such as cattle, horses, sheep, goats, swine, cats and dogs. Bovine babesiosis is one of the biggest concerns. The prevailing problems of the cattle industry are caused by *B. bovis* [5], *B. bigemina* [24] and *B. divergens* [25]. From an economic perspective, *B. bigemina* and *B. bovis* are recognized as the most problematic species, because they are distributed in places where their tick vectors are abundant. During bovine *Babesia* infection in young cattle (i.e., below 1 year of age) may not reveal symptoms, because they are resistant to *B. bovis* and *B. bigemina* acute disease [26]. In addition, individuals that are recovering from the acute babesiosis are potential reservoirs of the disease [27]. Signs of babesiosis differ among various *Babesia* spp., as they depend on their pathogenicity. Age, breed, immunological status of the animal and coinfections with other pathogens belong to factors determining bovine babesiosis infection [6]. Infection with *B. bovis* is severe and presents itself in high fever, ataxia, anorexia and circulatory shock. On the contrary, *B. bigemina* infections manifest more benignly but with more severe hemolytic anemia [28]. 

Babesiosis affects the livestock industry and animal and human health. Hence, a significant global economic loss in the beef and dairy industry is noticeable and worrying [29]. The main obstacles in the control of babesiosis are *Babesia* drug resistance and lack of effective vaccines. Current insufficient strategies for the control of the babesiosis include the use of ixodicides for vector control, controlled translocation of cattle, chemotherapy, chemoprophylaxis and selection of tick-resistant cattle [30]. The most recommended preventive measure against bovine babesiosis would be the immunization of cattle [31]. In this paper, we reviewed the current trends in the formation of an anti-babesiosis vaccine. Moreover, we analyzed the possible ways of developing one and summarize the achievements to date.

## 2. Advances in *B. microti* Vaccine Development

The recognition of vulnerable stages during *Babesia* spp. development both in final and intermediate hosts will enable the identification of biological processes that can serve as targets for a potential vaccine. It is important to consider the differences between *Babesia* species of different lineages, as they may differ in certain developmental features [19]. These may be crucial for vaccine development. Two major vaccine types have been identified in the available literature concerning the development of an anti-*B. microti* vaccine, namely, subunit and whole-pathogen vaccines. They have been further divided in terms of the nature of the specific antigens and parasites utilized in their development. 

### 2.1. Characterization of Whole-Parasite Vaccines

Whole-pathogen vaccines are the oldest but still a common method of providing immunity against many pathogens. Using inactivated or attenuated pathogens stimulates an immune response similar to the one observed in natural infections. Moreover, as whole pathogen vaccines contain all antigens of a given pathogen, the effect of antigenic variation and polymorphism is reduced. This gives them a major advantage over subunit vaccines [32]. In this section, the development of *B. microti* inactivated (killed) whole-parasite vaccines and attenuated whole-parasite vaccines are reviewed.

#### 2.1.1. Inactivated (Killed) Whole-Parasite Vaccines

In a *B. microti* murine model, intact RBC cellular membrane was shown to play an essential role in parasite-antigen-presenting cell (APC) interactions in the spleen and liver [33]. The efficacy of vaccines based on organisms contained within liposomal membranes, mimicking the structure of pRBCs, was demonstrated in a mouse model of malaria [34]. Liposomes offer both an attractive adjuvant, as well as a delivery system, for vaccines [35]. Furthermore, they can be readily lyophilized (freeze-dried), which is beneficial for vaccine storage and deployment. In a recent study, *B. microti*-killed parasites enclosed within liposomes that incorporated a mannosylated lipid core peptide were shown to induce protective immunity in mice with only low-level parasitemia. Lyophilization did not diminish vaccine potency when compared to a freshly prepared vaccine. Moreover, vaccination protected splenectomised mice, which suggests that it might be effective in asplenic individuals, one of the target groups of the future vaccine. Interestingly, vaccine-induced protection seemed to be dependent of CD4+ T cells and macrophage, with B cells and antiparasite antibodies showing little contribution [33]. Still, CD4+ T cells and macrophage-mediated response is insufficient to induce effective immunity; therefore, antibodies production stimulation is the most commonly proposed strategy for future vaccine development.

#### 2.1.2. Attenuated Whole-Parasite Vaccines

As attenuated whole-parasite vaccines, in contrast to inactivated ones, are based on weakened but still living pathogens, they might induce stronger and longer-lasting immunity. This approach has already been utilized in the control of related bovine babesiosis [36]. Seco-cyclopropyl pyrroloindole analog, tafuramycin-A, is molecule that is known to irreversibly bind with the parasite’s DNA and inhibit its replication. Mice vaccinated with three doses of tafuramycin-A attenuated *B. microti* pRBCs have developed protective immunity against homologous challenge with no parasitemia detectable through microscopy. It was determined that vaccine efficacy was dependent on intact pRBCs. Moreover, severe combined immunodeficient (SCID) mice intravenously injected with the vaccine did not develop signs of clinical babesiosis or parasitemia, implying its safety.

#### 2.1.3. Limitations of Whole-Parasite Vaccines

Although whole-parasite vaccines offer perspectives for future development of an anti-*B. microti* vaccine, they are characterized by several major disadvantages. A risk of contamination with adventitious agents is always associated with the in vitro manufacturing process of the *Babesia* parasites. Moreover, despite attempts at doing so [33,37], *B. microti* parasites appear impossible to be maintained in a continuous in vitro culture as a reliable source of the parasites. On the one hand, due to the fact of their relatively low immunogenicity, inactivated vaccines might require a human-specific strong adjuvant. On the other hand, more immunogenic attenuated vaccines may revert to virulence or lead to a residual virulence in immunocompromised individuals, one of the target groups of the future vaccine. So far, whole-parasite vaccines have not managed to elicit anti-*B. microti* antibodies of any clinical significance, which might potentially limit their efficacy. Moreover, attenuated vaccine efficiency was shown to be dependent on intact pRBCs [33]. This poses a challenge to the storage and deployment of such a vaccine. Furthermore, the presence of RBC membranes in whole-parasite vaccines implements a risk of inducing antibodies against a recipient’s RBCs [38].

#### 2.1.4. Further Development of Whole-Parasite Vaccines

It was proposed in a recent review, that the risks related to the inclusion of RBC membranes could be, to some extent, addressed through the use of human O negative blood to culture the parasites, as well as the reduction in the number of RBCs in the vaccine. The latter could be achieved by purification of parasitized RBCs from uninfected RBCs prior to the vaccine’s formulation. Moreover, dose-ranging studies aiming to identify the lowest parasite dose required for an effective vaccine could prove to be beneficial. Furthermore, a recently developed method for the purification of the *Plasmodium falciparum* parasites away from RBC membranes before formulation with an adjuvant could be adapted for *Babesia* inactivated vaccines [38,39].

The potential for a polyvalent vaccine against two major species, causing human babesiosis (i.e., *B. microti* and *B. divergens*) is also worth mentioning. It was determined that a lyophilized whole-parasite *B. divergens* liposomal vaccine could induce protection in mice against *B. microti* challenge, which is comparable to the one offered by a homologous *B. microti* vaccine [33]. Its efficacy against *B. divergens* was not tested due to the unavailability of a *B. divergens* murine model. As previously mentioned, the development of a reliable source of *B. microti* parasites is one of the major obstacles concerning whole-parasite vaccine production. *B. divergens*, in contrast to *B. microti*, can be cultured in vitro, which offers a potential solution to this problem [40]. Moreover, it was suggested in a recent review that *B. divergens* cell banks could be manufactured based on the protocols and methods used for *P. falciparum* cell banks. If successful, it would offer a dependable source of *B. divergens* parasites for future whole-parasite vaccine development [38,41,42].

### 2.2. Characterization of Subunit Vaccines

Subunit vaccines aim to induce the production of antibodies against specific protein-based antigens in order to facilitate pathogen elimination. After the introduction of *B. microti* sporozoites into the host’s bloodstream by a tick vector, only two developmental stages of *Babesia* development have been identified in the blood plasma–sporozoites and merozoites (Figure 1). As trophozoites exist exclusively within infected RBCs, they are, to an extent, shielded from the host’s antibodies. Sporozoites are present solely in the early stages of the infection, and subunit vaccines used against them reveal different efficacies, with protective antigens against only some *Babesia* species, which makes them both less desirable for a potential vaccine and difficult to study. This leaves *Babesia* merozoite as a prime candidate for the development of the subunit vaccine. Several approaches to eliminate *Babesia* parasites have been proposed, i.e., induction of antibodies against *B. microti* merozoite’s surface antigens, induction of antibodies against parasite’s internal antigens, induction of antibodies targeting the moving junction structure responsible for RBCs invasion, and induction of antibodies blocking parasite-induced cytoadherence of parasitized RBCs (pRBCs).

Surface antigens are excellent candidates for the subunit vaccine. Their presence on the parasite’s membrane provides an excellent access for both immune cells and antibodies. It also enables parasite’s clearance through opsonization-enhanced phagocytosis and complement pathway activation. 

*B. microti* surface antigen (BMSA) is a major surface antigen of *B. microti* merozoites, although its presence in trophozoites has also been shown [43]. It was determined, through confocal laser scanning microscopy and immunoelectron microscopy, that BMSA’s presence is mainly restricted to the parasite’s membrane, being secreted into the infected erythrocytes’ cytoplasm. Vaccination with BMSA was shown to have a significant inhibitory effect on the parasite invasion of the host erythrocytes and parasite growth in vivo, as well as a significant reduction in parasitemia in mice [43]. Moreover, a BMSA-specific monoclonal antibody was shown to control but not eliminate parasitemia in NOD/severe combined immunodeficiency (SCID) mice [43]. It is also worth noting that the post-vaccination upregulation of interleukins 17, 12p70, 10, 4 and 6 leads predominately to the Th2 immune response characterized by a vigorous expression of specific IgG1 antibody targeting the *Babesia* parasite [43].

*B. microti* surface antigen 1 (BmSA1) was identified through indirect immunofluorescence assay (IFA) followed by confocal laser microscopic observation as a secreted protein located in the parasites’ membrane and cytoplasm [44]. High fluorescence intensity in vivo indicated that the expression of BmSA1 was abundant and reached the highest values in the plasma merozoites. Furthermore, Western blot analysis followed by RBC binding assay revealed that recombinant BmSA1 (rBmSA1) adheres to mouse RBCs and, as such, could play a role in erythrocyte invasion [44]. An in vitro antiserum-neutralization test demonstrated that antiBmSA1 antiserum could inhibit parasitic growth. Moreover, anti-BmSA1 antibodies were shown to inhibit rBmSA1 adhesion to RBCs, which might explain one of the possible mechanisms of parasite growth inhibition. All these qualities make BmSA1 a promising candidate for a future subunit vaccine. However, it is important to note that when coupled with alhydrogel (i.e., an adjuvant approved for human use), the induction of high antibody titers did not affect the pathology of *Babesia* invasion in vaccinated mice [45].

BmSP44 is another surface antigen of *B. microti* merozoites. It was identified as a dominant reactive antigen and as a potential vaccine candidate by protein microarray screening [46]. Moreover, through IFA, it was established that BmSP44 is a secreted protein principally localized in the parasite’s cytoplasm. Mice immunized with BmSP44 demonstrated a significant difference in parasitemia when compared to the control group. Nevertheless, both vaccinated and control group mice still experienced high peak parasitemia [47].

Methionine aminopeptidases are enzymes involved in protein synthesis and the general metabolism of amino acids and proteins. Through indirect fluorescent antibody tests (IFAT) and confocal microscopy, it was established that *B. microti* methionine aminopeptidase protein 1 is mainly localized in the parasite’s cytosol. *B. microti* methionine aminopeptidase protein 1 was determined to elicit high titer antibodies, significantly lower peak parasitemia and facilitate earlier clearance of parasites in vaccinated mice [48].

*B. microti* heat shock protein 70 is an immunogenic protein identified through serological screening. It has been proven to elicit high antibody titers, as well as to significantly reduce peak parasitemia in immunized mice following challenge [49].

Profilin (PROF) is an actin-binding protein that plays an important role in dynamic turnover and reconstruction of the actin cytoskeleton. It is involved in locomotion and cell shape changes [50]. PROF was determined to be localized in the parasite’s cytosol by confocal laser microscopy. It is important to note that PROF has shown potential as a universal *Babesia* vaccine candidate. It was demonstrated that PROF is a common antigen among *B. bovis*, *B. bigemina* and *B. microti*. Moreover, PROF-based vaccines could all separately induce protective immunity against *B. microti* challenge in immunized mice. Small but significant reductions in peak parasitemia were observed in all groups, with BmPROF providing better protection than BboPROF and BbigPROF [51].

A moving junction is a multiprotein complex initially formed at the apical tip of apicomplexan parasites, such as *B. microti* and other *Babesia* species. It plays an essential role in RBCs invasion through the formation of a ring-like region of contact between surfaces of the invading parasite and host cell [52]. *B. microti* apical membrane antigen-1 (BmAMA-1) was shown by immunofluorescent microscopy to be localized in the cell’s surface and cytoplasm near the apical end of the parasite. Murine antibodies produced against the BmAMA-1 were determined to inhibit parasite growth in vitro, which suggests its role in RBC invasion [53]. A BmAMA-1-based vaccine was tested with the prime-boost strategy in a hamster model. AMA-1-specific antibodies were induced. However, animals did not exhibit significant protection against *B. microti* challenge [54].

*B. microti* rhoptry neck protein 2 (BmRON2) is another protein involved in the formation of the moving junction. It was identified through IFA to be localized in the apical end of *B. microti* merozoites. Antibodies against BmRON2 were shown to significantly inhibit parasite’s RBC invasion, further suggesting its role in this process [55]. BmAMA-1 and BmRON2 were tested as a potential vaccines in a hamster model of *B. microti*. Individuals immunized with a combination of BmAMA-1 and BmRON2 exhibited limited protection with a significant decrease in parasitemia and higher hematocrit values from day 6–10 post-challenge only. Interestingly, BmAMA-1 as well as BmRON2-based vaccines elicited comparable total amounts of antibodies to the combination vaccine but did not induce a significant protection. This suggests that the protection from the *B. microti* challenge is not only dependent on the antibody titer but also on specific epitopes targeted in both antigens [56]. 

Furthermore, N-terminal and C-terminal fragments of BmRON2 were evaluated as potential vaccines in a recent study. Both of them induced high specific antibody titers and reduced parasitemia following challenge. N-terminal BmRON2 fragment was shown to offer better protection [57].

*Babesia*-induced RBC cytoadherence to vascular endothelium is a phenomenon that is thought to contribute to the parasite’s persistence and virulence. If relevant for *B. microti*, a subunit vaccine based on parasite-derived RBC surface antigens could elicit antibodies to block parasitized RBC cytoadherence and facilitate their clearance in the spleen [38]. Although the role of this phenomenon in human babesiosis is unclear, it is well established for the related cattle parasite, i.e., *B. bovis* [58].

#### 2.2.1. Limitations of Protein-Based Subunit Vaccines

Protein-based subunit vaccines offer many promising perspectives for future development of a *B. microti* vaccine. They have many advantages such as no live components, so there is no risk of the vaccine triggering disease. They are relatively stable and suitable for people with a compromised immune system. However, several of their disadvantages have been identified in the available literature. The genetic variation in *B. microti*, isolated from the Northeast United States, was determined to be almost exclusively associated with genes encoding secreted and surface proteins [59]. A further study analyzed the 30 most seroreactive *B. microti* antigens through the use of the 41 published *B. microti* sequences [60]. It revealed that 18 of them were likely to be conserved, while 11 may be subjected to immune pressure based on their diversity [61]. This high diversity, likely to be driven by immune pressure, may limit the utility of secreted and surface proteins as future vaccine antigens. It is also important to mention the critical need for human-specific strong adjuvant for many developed candidate antigens, which causes further complications for the development of a *B. microti* subunit vaccine.

#### 2.2.2. Further Development of Protein-Based Subunit Vaccines

In recent years, many new strategies for the development of a *B. microti* subunit vaccine have been proposed and, accordingly, tools for further research have been developed. In order to identify new vaccine candidates, proteins derived from *B. microti* genomic expression libraries were tested for high immunoreactivity with sera from *B. microti*-infected humans or mice. With this approach, a novel protein, Bm2D41, was identified and tested as a potential vaccine. Bm2D41-immunized mice exhibited reduced peak parasitemia by 37.4%. However, it did not facilitate parasite clearance by day 30 post-challenge [62].

In a following study, through a similar approach, the top three most reactive antigens were selected: *B. microti* serine reactive antigen 1 (BmSERA1), *B. microti* Maltese cross form-related protein 1 (BmMCFPR1), and *B. microti* piroplasm β-strand domain 1 (BmPiβS1), in addition to the previously described *B. microti* alpha helical surface protein 1 (BmBAHCS1), for sensitivity and specificity studies. Sequence analyses performed with 38 *B. microti* isolates from the continental United States revealed minimal genetic diversity in BmSERA1, BmMCFPR1, and BmPiβS1, further increasing their attractiveness as potential vaccine candidates. Moreover, the cellular localization of BmSERA1, BmMCFPR1, and BmPiβS1 was shown, and a putative biological function was assigned to all four antigens, providing background for additional research [61].

In a recent study, a novel approach was developed [45]. It was based on a mammalian expression system that can produce a wide range of functional recombinant cell surface and secreted blood-stage *B. microti* proteins. Exactly 41 out of the 54 examined proteins were characterized based on antibody responses to parasite infections in human and mice hosts. They can be further evaluated for their suitability as candidates for the subunit vaccine in the future [45]. Moreover, through a combination of nanotechnology and mass spectrometry, a proteomic profile of *B. microti* was provided. Approximately 500 of the parasite’s proteins were identified with a broad array of functions assigned to them [63]. Glycosylphosphatidylinositol (GPI)-anchored, secreted, and transmembrane proteins are another antigen identified in the *B. microti* isolates from the Northeast United States, which can induce immune responses in vivo, and could be examined as candidates for a subunit vaccine [59].

Transfection, utilizing an electroporation method and genetic manipulation of *B. microti*, is the next tool proposed to aid in the vaccine development. A recent study has shown that obtaining viable transfected parasites under in vivo growing conditions is indeed possible. This method may expand our understanding of *B. microti* biology, host modulation and diverse parasite developmental stages through the utilization of reverse genetics and holds great potential for enhancing future vaccine development [64].

Immune checkpoints are signaling pathways known for their role in the downregulation of immune response. Programmed death-ligand 1 (PD-L1) interaction with programmed cell death protein 1 (PD-1) is known to decrease T-cell cytokine expression and impair its function. Pathogens exploit this pathway to escape elimination by the host’s immune system, which leads to their persistence. It has been shown that a blockage of this pathway enhanced the immune response against a related apicomplexan parasite, malaria, in a murine model. Recently, the BmPROF-PD-L1 vaccine was evaluated in a *B. microti* murine model. An in vitro study determined that vaccine-induced antibodies blocked PD-L1’s binding to PD-1. Moreover, immunization with BmPROF-PD-L1 led to decreased PD-1 expression in murine T cells and significantly reduced peak parasitemia post-challenge when compared with BmPROF-only vaccine [65]. This novel strategy of combining immune checkpoint inhibitory molecules with *B. microti* antigens offers a great potential for improving vaccine efficacy.

Trained immunity (TI) is a phenomenon characterized by the long-term functional modification of innate immune cells, which facilities more efficient responses to subsequent specific and nonspecific challenges. It was proposed in a recent review to utilize *Mycobacterium bovis* bacillus Calmette–Guerin (BCG) vaccine-induced TI in order to improve *B. bovis* and *B. microti* vaccine efficacy [66]. This hypothesis is supported by earlier studies that demonstrated that BCG inoculation protects mice against *B. microti* challenge [67,68]. Apart from strengthening innate immunity to control babesiosis, the BCG-induced TI could be utilized in auxiliary programs aimed at the eradication of the disease and in conjunction with *Babesia* spp. vaccines. Furthermore, the potential use of TI with recombinant BCG vaccines expressing *Babesia* immunogens offers an interesting new avenue for future *Babesia* spp. vaccine development [66]. 

## 3. Bovine Babesiosis Vaccines

Babesiosis is one of the most important arthropod-borne diseases of cattle and is very common in tropical and subtropical areas worldwide [69]. The economic loss is considerable, especially when animals without previous exposure are moved into an endemic area. Bovine babesiosis can be managed and treated, but the parasites are difficult to eradicate [70]. Cattle is the primary host and reservoir of *B. bovis* [70]. The major vectors of *B. bigemina* and *B. bovis* are *Rhipicephalus microplus* and *R. annulatus* but only in certain areas. *R. microplus* and *R. annulatus* are one-host ticks that complete their life cycle on a single host and preferentially feed on cattle. It can be transmitted transovarially [70]. 

*B. bovis* and *B. bigemina* are mainly found in tropical and subtropical regions. Although there are some differences in their distribution, these two organisms have been reported in Asia, Africa, the Middle East, Australia, Central and South America, parts of southern Europe and some islands in the Caribbean and South Pacific [70].

Babesiosis is characterized by fever, which can be relatively high, and varying degrees of hemolysis. Anemia has also been recognized as one of the symptoms and may develop rapidly. *B. bovis* can cause changes in red blood cells that result in their accumulation in capillaries, including those of the brain. Consequently, some neurological symptoms may arise (e.g., incoordination, teeth grinding, and manic behavior) and may cause or contribute to other serious syndromes such as respiratory distress [70]. It is also worth mentioning that animals after splenectomy develop a severe form of the disease, which results in death. The reason for this process is related to the fact that the spleen is responsible for destroying infected erythrocytes [71].

### 3.1. Vaccination Achievements in B. bovis Combat

Contemporary methods of fighting cattle babesiosis combine tick management, anti-*Babesia* drugs and vaccination with *Babesia*-attenuated strains [72]. It has been discovered that attenuated vaccines are not only effective but also reduce nervous symptoms, as well as reduce the transmission capacity of ticks [71,73,74,75]. The current strategy is a vaccination of less than one-year-old calves in endemic countries [72]. Conversely, the adult animals can develop acute babesiosis [71,76]. The process of production of live, attenuated vaccines involves preparation of a formula containing erythrocytes infected with *B. bovis* [71,77]. In most cases, the cells were obtained from infected splenectomised calves. Pathogen-free animals are infected with a proper parasitic strain and their blood is collected from the jugular vein during the acute phase of the reaction [3,71]. However, as Mazuz et al. pointed out, the currently available live attenuated blood-based vaccines present several drawbacks, such as the risk of the transmission of contaminating blood-borne pathogens and potential risks for reversion of virulence [72]. Furthermore, it is a challenge for supervisors of the process of vaccine production to keep the donors parasite-free, as the preparations are conducted in endemic regions [71]. Additionally, the vaccine requires cold chain to maintain its efficiency, which is a challenge in tropical, endemic regions [78].

The in vitro production of *Babesia* parasites has provided a novel strategy for vaccine development [71]. In vitro developed vaccines are not only safer, as they are less likely to transmit pathogens, but they also allow more controlled and standardized conditions [71,79]. Unfortunately, in vitro production requires a constant supply of erythrocytes and serum from donor animals, not to mention the adequate laboratory equipment and trained personnel [71]. It was observed that live *B. bovis* vaccines do not eliminate the parasite, but they rather produce disease-resistant carrier animals that may act as reservoirs for tick transmission [18,71,77].

#### Ideas for the Future of Vaccine Development against *B. bovis*

Scientists from all over the world are trying to discover the best possible way of fighting babesiosis among cattle. They are analyzing every stage of *B. bovis* development in order to find the most suitable antigens that will be targeted by the vaccine.

After being inserted into the cattle’s bloodstream, the parasite first invades erythrocytes. In the cell, *B. bovis* initiates its life cycle and eventually changes into merozoites that lyse the host cell and return to the blood vessel again. Now, it must re-enter a red blood cell in order to create gametocytes and finish the life cycle [71]. This process is possible due to the fact that the parasite has specialized GPI-anchor proteins called merozoite surface antigens (MSAs) 1 and 2 that allow binding to the cell membrane of erythrocytes [71,78,80]. Since MSAs are easily accessible, they are very likely to be a highly effective vaccine target [71]. Scientists, for example, Gimenez et al., claim that antigens against MSAs may block the invasion of blood cells and, consequently, protect the animal [78]. However, choosing proper domains of MSAs to maximize the efficacy of the vaccine still poses a problem that is being addressed by many teams worldwide [78,81,82]. Furthermore, it is suggested that treatment of GPI-anchor proteins with phospholipase C may reduce *B. bovis* merozoites’ ability for host cell invasion [18,83]. Another antigen candidate for the vaccine is ribosomal phosphoprotein P0. As Ramos et al. pointed out, this protein is found in all eukaryotic cells located on the surface. Anti-P0 antiserum inhibits the growth of *B. bovis* and blocks host cell invasion, and animals that underwent *B. bovis* infection display many anti-P0 antibodies; therefore, it is hypothesized that it would act as an efficient immunogen against bovine babesiosis [71,84].

After the initial contact, the parasite proceeds to produce proteins by specialized organelles of the merozoite apical complex: micronemes and rhoptries. These proteins are believed to play a crucial role in the host invasion. Therefore, they are profoundly analyzed by the scientists as potential vaccine candidates. The rhoptry-associated protein-1 (RAP-1) family is an example of these apical complex-associated proteins. They are associated with both surface of the merozoites, as well as with rhoptries, and are particularly interesting for scientists due to the fact of their high immunogenic potential [71,85]. Rodriguez et al. mentioned that cattle naturally or experimentally infected with *B. bovis* or *B. bigemina* mount RAP-1-specific antibodies that recognize merozoite surface-exposed epitopes [85]. Antibodies against RAP-1 inhibit the invasion of host’s erythrocytes [86]. High immunogenicity combined with increased antibody efficiency make anti-RAP-1 preparation a perfect vaccine candidate [85]. An example of micronemal protein is the thrombospondin-related anonymous protein (TRAP) family. Terkawi et al. describe them as proteins that are present in all apicomplexan parasites with conserved structures, consisting of a hydrophobic short N-terminal sequence, a von Willebrand factor A (vWFA) region, thrombospondin type 1 (TSP-1) domains, and a hydrophobic transmembrane sequence, followed by a short cytoplasmic tail. They probably function as adhesive molecules in cell-matrix interactions [87]. The study showed that in the asexual stage of *B. bovis*, the expression of BbTRAP2 is higher than other BbTRAPs. It has been discovered that the sera of animals previously infected with *B. bovis* contain a significant amount of anti-BbTRAP2 antibodies, which confirms that some regions of BbTRAP2 can function as potential vaccine targets [87]. Terkawi et al. proved that anti-BbTRAP2 IgG can not only neutralize the parasites and inhibit the in vitro growth of *B. bovis* but also prevent the invasion of merozoites [87].

While discussing the future of vaccines against bovine babesiosis, it is worth mentioning the transfection process. It is a laboratory technique of the incorporation of foreign nucleic acids into cells. This tool makes it possible to analyze gene functions and gene products in transfected cells [71,88]. Since the *B. bovis* genome has been sequenced, it is possible to use transfection methods in order to knock out specific genes or to express transgenes in *B. bovis*. The expression of foreign genes may be useful as a tool of co-expressing immunogens or introducing markers to identify vaccinated animals and to observe trafficking of the parasites in the environment, as well as their basic biology [71,89,90,91]. The transfection method creates countless possibilities for scientists worldwide not only to prevent infection of the parasite but also to fight against its vectors. A live attenuated strain’s genome can be integrated with numerous antigens that after immune stimulation during infestation can result in high levels of antibodies against incorporated antigen. The current research is mainly focused on creating dual vaccines protecting from the babesiosis as well as limiting the vector transmission. This could be achieved by the integration of tick antigens in the vaccine strains [71,72,92,93,94]. Transfection method of knocking out genes of the parasite genome also poses a great and promising solution in the fight against bovine babesiosis. Firstly, it allows scientists to analyze which genes are necessary in host–parasite interactions and, consequently, to discover efficient vaccine candidates. Secondly, reducing virulence factors may result in creating alternative and better-defined vaccine strains [71].

### 3.2. Vaccine Models for B. bigemina

The invention of vaccine against *B. bigemina* could be beneficial and could potentially decrease the prevalence of this protozoan. Several candidate molecules were identified in *B. bigemina*. 

#### 3.2.1. Protein-Based Vaccines

The BbiTRAP-1 (transmembrane protein) was characterized as a novel antigen, which contains adhesive domains, the von Willebrand factor A, thrombospondin type 1, and cytoplasmic C-terminus domains [95]. The BbiTRAP-1 (one of the three members of the TRAP family of proteins) predicted structure comprises a metal ion-dependent adhesion site that takes part in the interaction with the host cell. ELISA indicated that BbiTRAP-1 is not an immunodominant antigen. An in vitro study revealed that bovine antirecombinant BbiTRAP-1 antibodies can inhibit merozoite invasion. Interestingly, overall the predicted domain structures of BbiTRAP-1, 2 and 3 have a high level of conservation [95]. The other studies revealed the presence of the mic-1 gene and its product called micronemal protein 1. It contains a sialic acid binding domain, which enables it to invade host cells. In cattle naturally infected with *B. bigemina*, anti-MIC-1 antibodies were found, and 97.4% of the animals had antibodies against the MIC-1 A peptide. Antibodies recognizing the MIC-1 B peptide were detected in 83% of the sera. In vitro studies have also been performed. They showed 70% effectiveness in blocking merozoite invasion [96]. Another study demonstrated that one of the apical complex antigens of *B. bigemina*, rhoptry-associated protein-1 (RAP-1), in infected cattle induced the immunological response with INF-γ and RAP-1-specific IgG1 and IgG2 [97,98]. It was observed that RAP-1 elicits partial immunity in cattle [99]. In different studies, cattle were immunized with RAP-1. Native and recombinant forms of RAP-1 induced the T-cell response. This indicates that both forms of epitope could be used in vaccine production [85].

Another protein that was identified in *B. bigemina* is HAP2/GCS1. It is a specific sexual stage protein, and it takes part in membrane fusion during fertilization processes. In vitro studies were performed. The serum from rabbits (that were immunized with HAP2 peptide) was added to cultures with *B. bigemina*. The zygote formation of the parasite decreased in comparison to control cultures with an adjuvant. In four strains (that were geographically different), the sequences of the hap2 gene were analyzed. A high conservation at the amino acid level was observed. In the future, a better characterization of HAP-2 will be used in developing transmission-blocking vaccines (blocking the development of the parasite within the tick). Currently, there are no more studies describing the functions of HAP2/GCS1 in *B. bigemina* biology [99]. 

#### 3.2.2. Live Attenuated Vaccines

In another study, calves were vaccinated with a live attenuated vaccine produced using a modified micro-aerophilous stationary phase. The vaccine contained red blood cells infected with an attenuated parasite. After two months, the animals were intravenously infected with virulent *B. bigemina*. It was found that vaccination induced a sustained humoral response and an increased proliferation of CD4+ and CD8+ T cells. Exactly 12 calves were used and randomly divided into groups. Half of them were vaccinated and exposed, and none of the calves showed signs of babesiosis within 3 months. One animal died of intestinal causes, which was confirmed by necropsy. In addition to vaccinated animals, the study included two trials: a healthy control and an infected control. It was found that the level of CD4+ and CD8+ T cells decreased significantly in the group of infected animals compared to healthy calves. For this reason, the study suggests the importance of T-cell populations in acute bovine babesiosis [100]. 

#### 3.2.3. Transfection of *B. bigemina*

Silva and collaborators carried out a study using electroporation. They described the transfection of *B. bigemina*. The selectable marker taken to the research was blasticidin/blasticidin deaminase. The fusion gene consisted of green fluorescent protein-BSD (egfp-bsd) and elongation factor-1α (ef-1α). Furthermore, in homologous recombination, heterologous *B. bigemina* ef-1α sequences were also integrated into the genome of *B. bovis*. Knowledge about parasite transfection could be used in *Babesia* spp.-based vectored vaccines [101]. There are also studies that use the technique of the capillary feeding of ticks. TROSPA proteins involved in interactions between ticks and *B. bigemina* were selected. Female *R. microplus* was capillary fed with rabbit polyclonal antibodies against TROSPA. These antibodies were added to blood taken from cattle infected and uninfected with *B. bigemina*. In ticks fed with infected blood and anti-TROSPA antibodies, an 18% decrease in tick weight was observed, but no effect was shown on *B. bigemina* infection. There was no difference in mRNA levels of the TROSPA-encoding genes between ticks fed with infected and uninfected blood [102].

## 4. Remaining *Babesia* Species Vaccine Development

The other *Babesia* species being studied for vaccine development include *B. orientalis* and *B. divergens*. *B. orientalis* merozoite surface antigen 2c1 (BoMSA-2c1) is a member of the variable merozoite surface antigen (VMSA) family. This antigen is expressed only outside the RBCs and possibly allows the parasite to bind to the RBCs of the host. Maximum binding was achieved at an antigen concentration of 1.6 mg/mL. This process can be inhibited by specific anti-MSA-2c1 antibodies [103]. In turn, *B. divergens* homolog of AMA1 (BdAMA1) is an antigen that binds to the trypsin- and chymotrypsin-sensitive RCBs receptors. RT-PCR with degenerate primers was used to clone the BdAMA1-encoding gene. The purified IgG antibodies were obtained from the anti-AMA1 rabbit serum. Using anti-BdAMA1 antibodies, parasitemia was reduced by 50% [104]. El-Sayed et al. identified and characterized P0 protein (ribosomal P-protein) in *B. divergens*. They developed a recombinant P0 (BdP0) in *Escherichia coli*. Antiserum was taken from mice. A study revealed an interaction between anti-rBdP0 serum and corresponding legitimate parasite protein in cattle. It was confirmed by indirect fluorescent antibody tests and Western blotting. In addition, in a group of 68 bovine field samples, the immunogenicity of BdP0 was assessed by ELISA. The assay showed significant immunological reactivity in 19 positive samples of rBdpO lysate and 20 positive samples of *B. divergens* lysate. *B. divergens* was also cultured in vitro, and the growth was inhibited by anti-rBdP0 serum (*p* < 0.05). After six hours of incubation, merozoites had reduced by 59.88% the ability to invade bovine erythrocytes. rBdP0 is a potential vaccine candidate. In Europe, *B. divergens* is the major pathogen of human babesiosis, so performing more studies is crucial [105].

## 5. Summary

This review summarizes information on vaccines against *Babesia* species, specifically *B. microti*, *B. bovis*, *B. bigemina*, *B. divergens*, and *B. orientalis*. Molecular candidates for the production of vaccines have been presented. The list of candidates includes BMSA, BmSA1, BmSP44, *B. microti* heat shock protein 70, BmPROF, BboPROF, BbigPROF, BmAMA-1, BmRON2, N-terminal and C-terminal fragments of BmRON2 for *B. microti*, MSA, ribosomal phosphoprotein P0, RAP-1, BbTRAP2 for *B. bovis*, BbiTRAP-1, MIC-1, RAP-1, HAP2/GCS1, TROSPA for *B. bigemina*, BoMSA-2c1 for *B. orientalis,* BdAMA1 and BdP0 for *B. divergens* (Table 1).

Protein-based subunit vaccines and whole-parasite vaccines against *B. microti* were described. Several of their advantages, as well as disadvantages, were pointed out. Trials to use an attenuated vaccine for *B. bovis* and *B. bigemina* were also reported. The process of transfection was also proposed as a method to fight against babesiosis. The gathered conclusions suggest significant progress in the development of a vaccine against *Babesia* spp. and, at the same time, show the need for further research.

## Figures and Tables

**Figure 1 pathogens-12-00300-f001:**
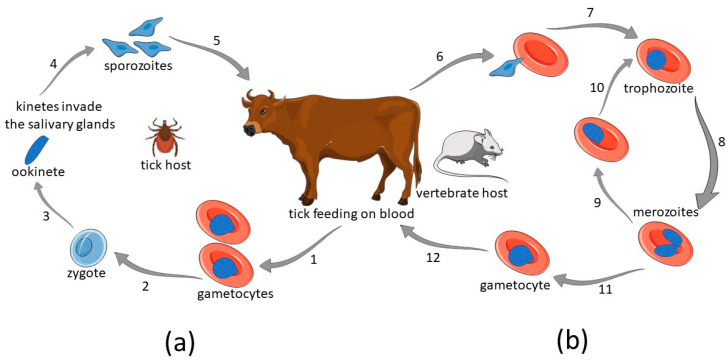
The life cycle of *Babesia* spp. (**a**) Sexual cycle in the definitive tick host. 1. A tick feeds on blood and ingests *Babesia* spp. gametocytes. 2. In a gut of a tick, gametocytes differentiate into gametes which unite forming a zygote. 3. Ookinetes are formed and then kinetes enter the salivary glands of a tick. Transovarial transmission in a tick can also occur. 4. Invasive sporozoites are produced. 5. A tick transmits sporozoites while feeding on vertebrate host. (**b**) Asexual cycle in the intermediate host. 6. Sporozoites invade erythrocytes and undergo asexual reproduction. 7. Sporozoites develop into trophozoites. 8. Trophozoites divide into merozoites. 9. Merozoites spread throughout the blood. 10. Merozoite develops into trophozoite. 11. Some of the merozoites produce male and female gametocytes. 12. Gametocytes in the intermediate host’s red blood cells.

**Table 1 pathogens-12-00300-t001:** Molecular candidates for the production of *Babesia* vaccines.

Species	Protection for Animals	Protection for Humans
*B. microti*	X	BMSA BmSA1 BmSP44 *Babesia microti* methionine aminopeptidase protein 1 *B. microti* heat-shock protein-70 BmPROF BboPROF BbigPROF BmAMA-1 BmRON2 N-terminal and C-terminal fragments of BmRON2 Bm2D41 BmSERA1 BmMCFPR1 BmPiβS1 BmBAHCS1
*B. bovis*	MSA-1/MSA-2 PHOSPHOPROTEIN P0 RAP-1 BpTRAP2	BboPROF
*B. bigemina*	BbiTRAP-1 MIC-1 RAP-1 HAP2/GCS1 TROSPA	BbigPROF
*B. divergens*	BdAMA1 BdP0	BdAMA1 BdP0
*B. orientalis*	BoMSA-2c1	X

## Data Availability

Not applicable.

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
