# Peer review of "Advances in Babesia Vaccine Development: An Overview"

_pathogens, 2023, doi:10.3390/pathogens12020300_

Round 1

Reviewer 1 Report

The review summarizes studies conducted in pursuit of effective vaccines against babesiosis. The information is relevant for a broad audience who are interested in human and non-human animal health.

Figure 1 is very simple and the legend doesn’t explain the figure. A more detailed figure would be useful for readers. For example, include infection of midgut, development of kinetes with infection of ovaries, if applicable, and salivary glands in the tick host.

The paper will be improved by adding more discussion about why antibody stimulation is the prime goal of vaccination, particularly as they point out in lines 290-291 and 306-307 that B-cells and antibodies were shown to have little effect on protective responses, at least in the study cited.

References are needed for paragraphs in lines 93-117.

Add RBC abbreviation at first use, line 45.

Line 118-120 is an incomplete sentence.

Line 79, “most proper” implies that other control measures shouldn’t be used or are minimal considerations. A different way to state the importance of vaccination should be used.

Line 68, “breeds” should be used instead of “races” for cattle.

Line 76, What does drug resistance refer to, parasites or ticks?

Reviewer 2 Report

Dear Authors, 

I have some minor suggestions for the authors to improve their work.

The manuscript is interesting and well-designed; a grammatical revision is required throughout the text; for example, some articles, commas, adverbs, etc., must be included.

Some significant points should be addressed before the manuscript can be considered for publication.

- Throughout the text, check the conjugation of the verbs.

- The paper needs to be checked carefully for typos and grammatical errors.

- The scientific names and Latin names should be written in italics.

- In the reference list, some names of species are not italics ((even in the text, check this aspect). - Keep the same font style and size throughout the manuscript.

The review is interesting and important for identifying the vaccine advances in babesiosis. However, the manuscript is partially difficult to read, and I would propose a re-organization of the manuscript (For example, in Table 1, a column could be added with the data on use and concentrations that protect animals or humans); there is information. The main emphasis should be on relevant examples. 

Round 2

Reviewer 1 Report

The author's have addressed my comments.

Author Response

Dear Reviewer,

we are grateful for the approval of the corrections that were made in the manuscript.  We would like to thank for the improved rating of the text.